# A Method of Supplementing Reviews to Less-Known Tourist Spots Using Geotagged Tweets

**Victor Silaa** [1,*,†] , **Fumito Masui** [2,†] **and Michal Ptaszynski** [2]

1   Graduate School of Engineering, Kitami Institute of Technology, Kitami 090-8507, Japan
2   Department of Computer Science, Kitami Institute of Technology, Kitami 090-8507, Japan;
    f-masui@mail.kitami-it.ac.jp (F.M.); michal@mail.kitami-it.ac.jp (M.P.);
*   Correspondence: d2071308070@std.kitami-it.ac.jp
†   These authors contributed equally to this work.

**Abstract:** When planning a travel or an adventure, sightseers increasingly rely on opinions posted on the Internet tourism related websites, such as TripAdvisor, Booking.com or Expedia. Unfortunately, beautiful, yet less-known places and rarely visited sightspots often do not accumulate sufficient number of valuable opinions on such websites. On the other hand, users often post their opinions on casual social media services, such as Facebook, Instagram or Twitter. Therefore, in this study, we develop a system for supplementing insufficient number of Internet opinions available for sightspots with tweets containing opinions of such sightspots, with a specific focus on wildlife sightspots. To do that, we develop an approach consisting of a system (PSRS) for wildlife sightspots and propose a method for verifying collected geotagged tweets and using them as on-spot reviews. Tweets that contain geolocation information are considered geotagged and therefore treated as possible tourist on-spot reviews. The main challenge, however, is to confirm the authenticity of the extracted tweets. Our method includes the use of location clustering and classification techniques. Specifically, extracted geotagged tweets are clustered by using location information and then annotated taking into consideration specific features applied to machine learning-based classification techniques. As for the machine learning (ML) algorithms, we adopt a fine-tuned transformer neural network-based BERT model which implements the information of token context orientation. The BERT model achieved a higher F-score of 0.936, suggesting that applying a state-of-the-art deep learning-based approach had a significant impact on solving this task. The extracted tweets and annotated scores are then mapped on the designed Park Supplementary Review System (PSRS) as supplementary reviews for travelers seeking additional information about the related sightseeing spots.

**Keywords:** less-known tourist spots; on-spot reviews; POI; PSRS; BERT

## 1. Introduction

In this paper, we present our study in developing a system for tourists review collection and visualization that accommodates on-spot reviews for less-known tourist spots. On-spot reviews are online opinions assumed to be posted at the target facility. In this study, we apply geotagged tweets as potential on-spot reviews, estimate their adequateness as reviews and further apply the verified tweets as on-spot reviews in the designed Park Supplementary Review System (PSRS).

Recently, there has been a rapidly increasing demand for the application of information technologies in the field of tourism (defined with a blanket term of Tourism Informatics). Diverse Big Data have been applied to tourism research and have made considerable improvements, for example, in the development of recommendation systems (Masui et al. [1]), navigation systems (Yoshida et al. [2]), and regional content tourism support systems (Masui et al. [3]). The main goal is to promote tourism of a specific place and to provide personalized information as per specific search. Apart from the developed systems, the task

of analyzing tourism information is of great importance. It enables the collection of large amounts of data to supplement the developed systems. By data sources, tourism-related Big Data generally fall into a few broad categories, which include the following.

- User Generated Contents (UGC), defined as data generated by users which includes online textual and photo data, etc.;
- Device Data (generated by devices), which includes GPS data, roaming data from mobile devices, Bluetooth data, etc.;
- Transaction Data (generated by operations), with the likes of Web search data, Web page visiting data, or online booking data.

These carry different information and different data types which may address different tourism issues as explained by Ling et al. [4].

The Internet today has vastly altered the data landscape, by accumulating a lot of information. People, businesses, and devices have all become data factories that are pumping out large amounts of information to the Web each day, Askitasklaus et al. [5]. This huge amount of data shared on the Internet can be utilized to foster tourism activities in a given specific area. Internet users can easily express their opinions about a product, service or a place they have recently visited using popular Social Networking Services (SNS), such as Twitter, Facebook, or Instagram and reach millions of other potential visitors. In this way, people tend to transmit their daily events in the form of diaries and textual messages using online social services such as blogs, online posts, microblogs, and other SNS. Among many SNS, the one that has been greatly popular for people to express their opinions, share their thoughts, and report real-time events has been Twitter (https://twitter.com/, accessed on 15 January 2022). Many companies and organizations have been interested in utilizing the data appearing on Twitter to study the opinions of people towards different products, services, facilities, and events taking place around the world. Through Twitter, a great number of messages (known as "tweets") are posted daily because of its simplicity. Moreover, with GPS technology implemented in mobile phones and computers, sightseers as well share their views and pictures regarding their tour experiences on Twitter. This type of information is valuable and important in facilitating tourism activities of the specific area tagged with GPS information. Online opinions thus can have a great impact on brand, product or place reputation. For this reason, some potential visitors make informed decisions based on online opinions. Primarily, there is a number of online review sites for tourism related activities, such as TripAdvisor (https://tripadvisor.com/, accessed on 15 January 2022), Booking.com, or Expedia (https://www.expedia.com, accessed on 15 January 2022).

Unfortunately, less-known and rarely visited sightspots often do not accumulate sufficient number of valuable opinions. Therefore, to address this, we introduce the concept of using on-spot reviews (on-spot tweets with contents verified to contain visitor opinions). These are Internet opinions about the target spot extracted from geotagged tweets. To prove the adequateness of the extracted information we propose our classification method that uses a fine-tuned BERT model. Previously, Shimada et al. [6] introduced a method to identify on-site likelihood of tweets using a two-stage method, a rule based and contextual approach. Unlike them, in our proposed method we prove adequateness using a fine-tuned BERT model.

Approved geotagged tweets are mapped as on-spot reviews in the designed system (PSRS). This is realized as efforts to cultivate newly Point Of Interest (POI) and to supplement additional information to the less-known places in the target spot (Serengeti and Ngorongoro) National Park (NP), which are famous and largest NP in northern Tanzania. Serengeti's annual great wildebeest migration is an iconic feature of the park which is happening around the end of year. The two parks are in the list of UNESCO World Heritage Sites with Serengeti NP property changing seamlessly to Ngorongoro Conservation Unit (see Figure 1 for details). The plains of Serengeti NP, comprising 1.5 million hectares of savanna, while the annual migration of two million wildebeests, with thousands of other ungulates in search of pasture and water, engage in a 1000 km long annual circular trek

spanning the two adjacent countries of Kenya and Tanzania. It is known to be one of the nature's most impressive spectacles (https://whc.unesco.org/en/list/156/, accessed on 15 January 2022). The two spots together cover the area of more than twenty thousand square kilometers with many sightspots scattered around the area. Because of its wide area, some spots are less-known among sightseers than others and therefore rarely visited, thus accumulating few reviews.

Additionally, the wildebeest migration is a famous but seasonal scenery across the target spot. Precise timing is entirely dependent upon the rainfall patterns each year. Hence, POI also differ periodically. Despite the fact that the migration and animal spot can be predicted, in this study, we take extra efforts to cultivate new POI pointed out in tweets by tourists. This is an important task as it can improve tourism activities of those target spots. Moreover, if the method is verified as effective, it can be applied also to other such attractive, yet not often visited sightseeing spots, all around the globe, in any country.

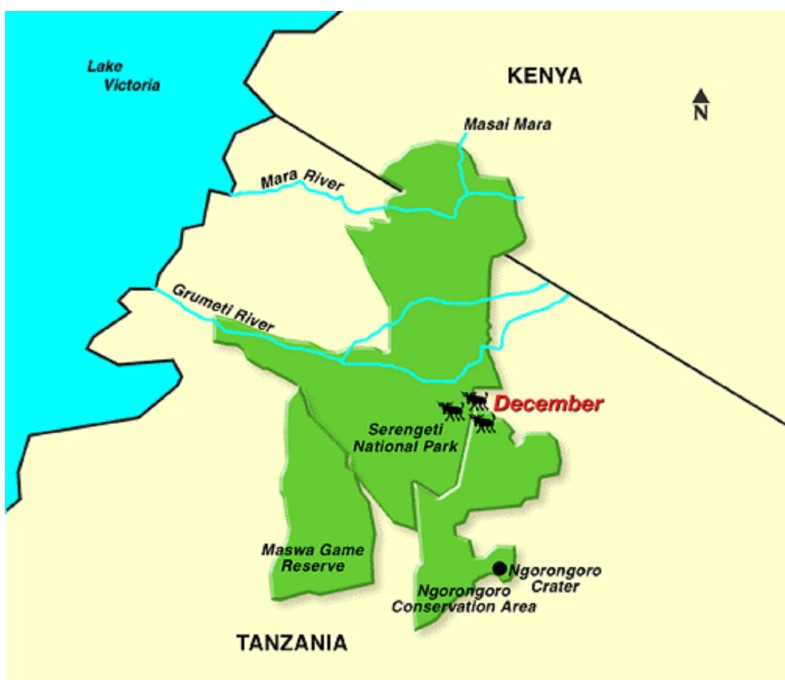

**Figure 1.** A map and a bird's eye view on the target sightseeing spots analyzed in this study—Ngorongoro and Serengeti NP.

Therefore, in this study, we propose a method of obtaining tourist on-spot reviews from the Internet to complement the least reviewed sightspots by extracting information directly from geotagged tweets. Tweets are considered geotagged if they include geolocation information assigned to it. We treat tweets that include the name of the target spot as potential tourist on-spot reviews. Results published in this paper represent an effort to complement reviews information for less-known places and rarely visited sightspots areas. Therefore, this article, by presenting a method to support less-known, yet valuable tourist attractions by cultivating on-spot reviews automatically with automatically collected and analyzed geotagged tweets, presents an important contribution for Tourism Informatics in general. The main scientific problem we solve in this paper is answering the question of how to identify the authenticity and utility of the extracted tweets as equivalents of online reviews.

Various types of approaches were developed and improved to tackle the task of extracting valuable information from the Internet by proposing POI recommendations that provide a location suitable as per user's preferences. Some of the most successful approaches so far include rule-based or statistical approaches, while novel Deep Learning-based approaches are yet to be commonly used (Minaee et al. [7]).

This study attempts to address the task of obtaining online reviews (UGC data source category) by extracting Twitter microblog posts (tweets), in form of textual data with the aim to extract useful information and further create a classifier to determine whether the tweets are likely to carry similar information. This task is widely recognized as text classification which is one of the fundamental tasks in Natural Language Processing (NLP).

Due to its nature, text classification has important implications for NLP tasks, which aim to either analyze, understand, or produce human language. Text classification has a large potential for various applications in the domain of text mining, especially those that require semantic analysis, such as author profiling and sentiment analysis (Sboev et al. [8]).

Categorizing tweets has been challenging due to insufficient contextual information and noisy possession. Recently, Zahera et al. [9] suggested a disaster management multi model approach for identifying actionable information from disaster-related tweets using Bert, graph attention network and relation network.In their work, the focus is on multiple classification so as to allow rapid detection of various categories of tweets. Their approaches outperform state-of-the-art approaches. Masaki et al. [10] proposed a real-time analysis method of detecting tourists spots from geotagged tweets using location information from tweets and a time-series changes. Their method revealed improvements compared to their previous moving-average method. Compared to above related works, we use BERT for both binary text classification (on-spot tweets or not) and multi text classification where we identify the semantic polarity of the tweets using a three and five stage rating score.

The main contribution of this study is four-fold. Firstly, this study proposes a mining framework that cultivates on-spot reviews and related POI from geotagged tweets by using location clustering and BERT neural network model. Secondly, it adds most probable rating score to the on-spot reviews extracted by learning the sentiment orientation of the tweets using BERT neural network model. Thirdly, we develop a corpus of on-spot annotated tweets which can further be applicable in other NLP tasks. Finally, we designed a web system (PSRS) and use selected and rated tweets as touristic information.

In a global perspective, this study intends to support the local tourism sector in Tanzania specifically in the area of wildlife-based tourism as one of the promising and fastest-growing sector among others in Tanzania, with the selected target spot attracting the most sightseers [11–13].

The rest of this paper is organized as follows. Section 2 introduces related works and previous research. Section 3 briefly outlines the proposed method used in this research. Section 4 describes the applied data. Experiment setup and analysis of the results are discussed in Section 5. Additionally, Section 6 discusses results and various experimental findings. Section 7 introduces the designed system (PSRS). Finally, in Section 8 we present conclusions and future works.

## 2. Related Work

This section presents some of the most relevant previous studies related to ours to provide a thorough overview of the topic. Specifically, we review earlier works with rule-based approaches to tourist information extraction, and summarize the most important findings from previous studies.

### Extraction and Presentation of Tourism Information

In recent years, various studies have been conducted on the provision and analysis of tourism-related information on the Web.

Okamura et al. [14], proposed an automatic score generation method in favor of the least reviewed local restaurants by analyzing the reviews posted on the Internet. They proposed a decision model using a convolution neural network with two hidden layers under a back propagation algorithm.

Lee et al. [15] proposed a geo-social event detection system by monitoring crowd behaviors indirectly through Twitter. Their proposed method focuses on temporal features within the target spot as an important factor for extracting geo-social events.

On the other hand, Cheng et al. [16] proposed a method of predicting a user's location by focusing on the content of the tweet. Their method relies on the approach of the three key features which are (a) reliance purely on tweet content; (b) classification of words in tweets with a strong local geo-scope; and (c) a lattice-based neighborhood smoothing model.

Sakaki et al. [17] studied event detection from Twitter data, by applying Kalman filtering and particle filtering, which are widely used for location estimation in pervasive computing.

In summary, these studies show that User-Generated Content has become a popular medium for expressing opinions and sharing knowledge about items such as products and travel entities while on the other hand, an essential tool for researchers to extract information.

Tourist Information Recommendation

Several studies propose recommendations of POI by suggesting suitable locations based on user preferences.

Oku et al. [18] proposed a method of mapping geotagged tweets to sightspots based on the substantial activity regions of the spots. Their method learns from One-Class Support Vector Machines-based classifier which first extracts temporal and phrasal features of the pattern sentences for classification and further maps the tweets into respective regions. Location-based SNS such as Foursquare were useful in this study by providing geotagged post data.

Shimada et al. [6] suggested a method that identifies on-site likelihood adequateness of posted tweets with a two-stage method, which includes rule-based filtering, and a machine learning (ML) technique. In their method, a previous and next tweet was taken into consideration as a potential target defining context information. The analysis of the experimental results shows the effectiveness of the combined applied techniques.

Overall, as discussed above, there have been some studies attempting to extract characteristics of the target regions based on geotagged contents.

However, while many of the above-mentioned studies, focus on the extraction of information using either rule-based approaches or simple ML classifiers (e.g., SVM), we focus on extraction of online opinions and assigning scores by adopting a state-of-the-art neural network-based architecture (BERT).

## 3. Proposed Method

In this section, we describe the proposed method that:

(i)     classifies on-spot tweets from Twitter data by incorporating clustering and BERT, and
(ii)     adds rating information to on-spot judged tweets

In this section, we firstly, introduce the procedures involved in realization of the proposed method and further discuss its inner processes at each stage.

The proposed method incorporates location clustering and classification techniques. The outline of the procedures involved, consists of a series of stages as observed in Figure 2.

Figure 2 outlines the procedures involved in the realization of the proposed method. In stage A, tweets are collected from the Internet by specifying the keywords "ngorongoro" and "serengeti", which may appear anywhere in the tweet, by using an accredited Twitter API (https://developer.twitter.com/en/products/twitter-api, accessed on 15 January 2022). In stage B, we cluster the collected tweets by location. A K-means algorithm, which is a vector quantization algorithm introduced by Hartigan et al. [19] is applied to tweets' location information to automatically partition them into clusters K, by calculating the nearest mean from cluster centroid. Tweets located within the target spot estimated boundaries are retained. Since the target spot boundaries are not explicitly specified, we decide our target spot boundaries with the help of Google maps (https://maps.google.com/, accessed on 15 January 2022) which highlights the East, West, North and South boundaries of the target spot as follows;

- East = 2°24′13.5″ S 35°16′03.4″ E
- West = 2°11′27.2″ S 34°07′58.8″ E
- North = 1°26′33.6″ S 34°48′45.0″ E
- South = 3°11′02.6″ S 34°38′08.2″ E

In stage C, we manual annotate location clustered tweets as either on-spot or not. We also assign sentiment score to the tweets. To accomplish this task, we use three annotators. The details of annotation task is discussed in details in later part of this article.

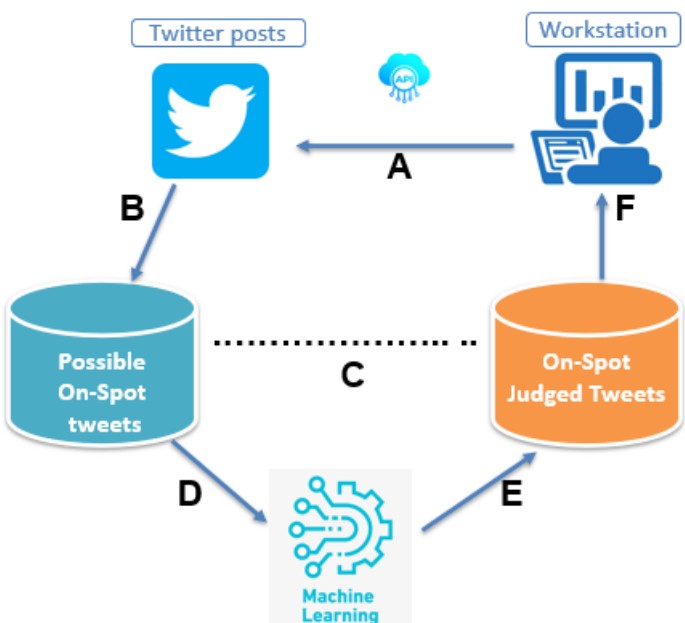

**Figure 2.** Outline of procedures constituting the proposed method.

In stage D and E, we trained our classifier to predict tweets and the sentiment score assigned to them and further evaluate the model performance. We adopt a pre-trained BERT neural network model for this task. In stage F, we map selected and rated tweets as touristic information in the designed system (PSRS).

### 3.1. Location Clustering of Tweets

Clustering is the task of grouping a set of objects in such a way that objects in the same group (called a cluster) are more similar (in some sense) to each other than to those in other groups (clusters).

Using K-means clustering, the number of clusters must be decided beforehand. Based on collected tweets data distribution, we adopt a technical approach method to identify the optimal number of clusters using an Elbow method, Average Silhouette method, and Gap statistics method, respectively. Figure 3 shows the results of the most optimal number of cluster groups as obtained from an Elbow method.We can further observe a 2D representation of the obtained clusters with the distribution of extracted tweets as shown in Figure 4.

We analyze the results of location clustering and consider tweets within the target spot boundaries as potential on-spot reviews. We use filtering approach to distinguish tweets beyond target spot boundaries. Table 1 shows few examples of on-spot judged tweets. In the next procedure, we identify on-spot tweets and assign sentiment scores to them by manual annotation, putting into consideration sets of features established and discussed in the following section.

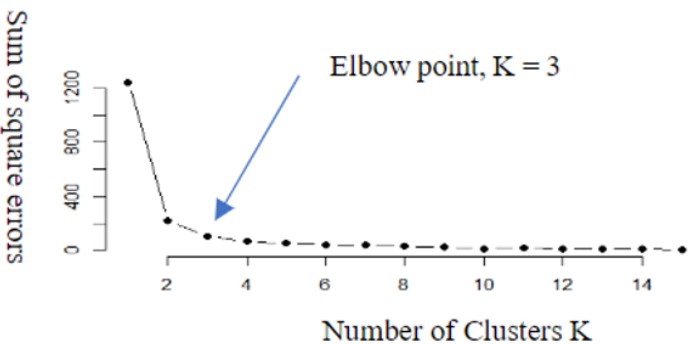

**Figure 3.** Number of clusters (K).

**Table 1.** Examples of location-clustered tweets.

| Tweet Contents | Longitude | Latitude | Cluster |
| --- | --- | --- | --- |
| ngorongoro crater hippo pool @ ngorongoro crater | 35.6762 | −3.1540 | 2 |
| successfuly completed a baloon safari in central serengeti | 36.6833 | −3.3666 | 2 |
| serengeti sunset through the trees as seen from the place we stay | 34.5902 | −2.1469 | 2 |

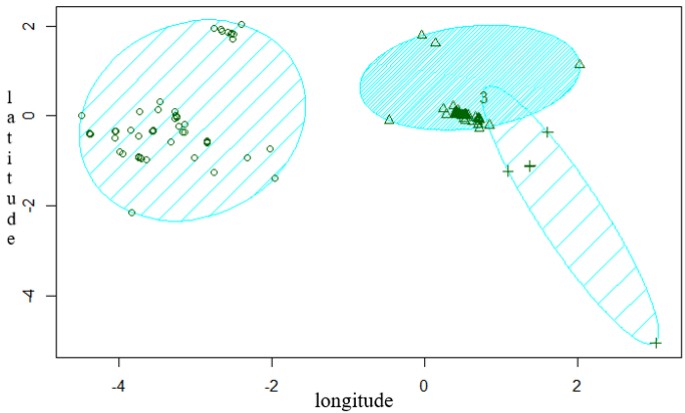

**Figure 4.** Clusters of geotagged tweets.

*3.2. Corpus Annotation*

Annotation is a methodology for adding information to a document at some level, such as a word, a phrase, paragraph, section, or the entire document. Manual text annotation is an essential part of text analytics. Although annotators (workers performing the manual annotation) work with limited parts of data sets, their results are applied to further train automated text classification techniques and thus affect the final classification results. Automated text analytics methods rely on manually annotated data by building their heuristic, or statistical rules, or neural networks on such annotated data (Bobicev et al. [20]). In the annotation process, we define the text to annotate, set labels to put in tweets, and we discard tweets with a certain degree of ambiguity so as to reduce noise when classifying.

To accomplish this task, we asked three annotators to carefully assign the clustered 1273 tweets. Additionally, annotators also assigned sentiment score of tweets as either positive, neutral or negative. Table 2 highlights the summary of annotated tweets (here referred to as corpus of annotated tweets).

**Table 2.** A summary of annotated tweets (corpus).

| Annotated Tweets | Unit |
|---|---|
| Sample of all tweets | 1273 |
| Sample of score assigned tweets | 1273 |
| Sample of On-spot annotated tweets | 974 |
| Sample of Not On-spot annotated tweets | 299 |
| Avg. length (char) of all tweets | 118 |
| Avg. length (words) of all tweets | 20 |

Table 3, shows a number of examples of the annotated tweets. "1" indicates an on-spot annotated tweet, while "0" indicates a not on-spot tweet. The remarks column indicates a reason for such annotation. In Table 3 for example, a tweet "k" is not tweet from the target spot however the name of target spot was tagged in it. For this reason, it is important to manually annotate our data.

**Table 3.** Examples of annotation made from location-clustered tweets.

| id | Tweet | Label | Remarks |
|---|---|---|---|
| i | i'm at serengeti national park | 1 | at target spot |
| j | I'm at ngorongoro wildlife lodge in ngorongoro | 1 | at target spot |
| k | I'm at serengeti park in hodenhagen niedersachisen | 0 | different spot |
| l | forbookingsafariserengeti, ngorongoro, mikumi national park | 0 | advert |

*3.3. Inter-Rater Agreement*

The reliability of annotations and adequacy of assigned labels are especially important in the case of sentiment annotations. In particular, Plaban et al. [21], addressed the importance of evaluating the reliability between annotators for statistical accuracy. To measure the agreement between three raters, we use Cohen's kappa coefficient, Cohen et al. [22].

Kappa coefficient between two or more annotators can be computed by using the following formula:

$$\kappa = 1 - \frac{1 - P_o}{1 - P_h} \tag{1}$$

In this above equation, $P_o$ is the relative observed agreement among raters, and $P_h$ is the hypothetical probability of chance agreement, using the observed tweets data to calculate the probabilities of each observer randomly seeing each category.

When kappa = 1, the annotators are in complete agreement. When the score is negative, it shows that there is no effective agreement between annotators, or the agreement is worse than random.

In addition, the hypothetical probability of the chance of agreement can be computed using the following formula:

$$P_h = \frac{1}{N^2} \sum_k n_{k1} n_{k2} \tag{2}$$

where $k$ represents categories, and $N$ being the number of observations to categorize. In this study, the degree of agreement between the three annotators was calculated as 0.37. Kappa's have specific interpretations, and 0.37 can be interpreted as "substantial", "fair", "medium" or "somewhat good" depending on the interpretation (Landis and Koch et al. [23]). This value, however, is not high to say annotators have an agreement on the annotation results. From this observation, we can assume that the final results of our proposed model was also affected by the low level of agreement between annotators. One way to improve this is by

carefully removing ambiguous tweets, which will be our improvement consideration in our future work.

### 3.4. Feature Selection

Many tourism-related tweets on Twitter do not contain on-spot information. One of the solutions to extract on-spot tweets is by classifying them as such by using a machine learning-based classifier. In collecting tourists' tweets, it is necessary to determine the conditions of considering which tweets are tourists' tweets. Therefore, we introduce a set of tweets classification features to be used for the automatic classification as follows:

Tweet location: We observed that tweets tweeted within the radius of the target spot's boundaries (latitude and longitude) introduced in the previous section which was acquired using Google's Geocoding API (https://developers.google.com/maps/documentation/geocoding/overview, accessed on 15 January 2022) often had a high chance of becoming a valuable on-spot review.

Presence of "NOW": The word "now" is a characteristic keyword on Twitter. Although the presence of the word does not always indicate on-spot information, it is considered to suggest a high probability of the tweet containing on-spot information. We, therefore, retain tweets with this word.

Presence of a mention "@ Target spot": In many cases, tourists' tweets about places they are sightseeing are accompanied with images the users attach to tweets by using mobile camera functions. At that time, expressions such as "@ Serengeti national park" frequently indicate places visited after "@".

Bag of Words (BOW): All words from the whole corpus with the term frequency for the BOW language model, which contains 1273 sentences.

### 3.5. BERT for Classification

We adopted a BERT model for the training and evaluation of our classifier. BERT architecture is defined as follows; "BERT stands for Bidirectional Encoder Representations from Transformers. It is designed to pre-train deep bidirectional representations from an unlabeled text by jointly conditioning on both the left and right context. As a result, the pre-trained BERT model can be fine-tuned with just one additional output layer to create state-of-the-art models for a wide range of NLP tasks" [24]. The Transformers architecture is the main block in BERT. Transformers is a deep learning model used primarily in the field of NLP. It is deeply bidirectional which means it learns from both sides during the training phase. Its token input representation is constructed by summing the token, segment, and position embeddings [25]. One of the biggest challenges in NLP is the shortage of training data. However, by adopting a fine-tuned BERT model that takes into account the context orientation of the token in the sentence, it is in theory possible to obtain high results with only a limited amount of training data. This is the main reason behind adopting this approach. This advantage is due to the impact of the pre-training mechanism, which established the formula of transfer learning in NLP. The transfer learning process in NLP can be achieved with two major processes, namely, a pre-training process and a fine-tuning process.

## 4. Applied Data

As training data for the proposed ML classifier, in this study, we used tweets collected from Twitter (see Figure 5 for an example of geotagged tweets collected).

{"created_at": "2019-08-13 10:08:30",
"text": "He crossed the savanna to find the sunset in her eyes\u2026 https://t.co/h14wFmACCP"
"geo" : {34.854, -6.307}

{"created_at": "2020-05-2 2:02:24",
"text": "Honeymoonin' in the Serengeti! We can't think of a better way to enjoy a honeymoon!
 What do you think?… https://t.co/nFQ1hDFyq5",
"geo" : {34.711476, -2.17650617}

{"created_at": "2020-10-2 6:44:40",
"text": "Jambo from Serengeti! @ Serengeti National Park https://t.co/CwgdookCJG",
"geo" : {34.84190712, -2.33646445} |

**Figure 5.** Examples of geotagged tweets.

The tweets used in this study were collected within a period of eight months, from June 2019 through February 2020. The data was collected from Twitter by searching for the keywords "ngorongoro" and "serengeti" which could occur anywhere in the tweet that was finally included in the dataset. Recently, due to the coronavirus outbreak, wildlife tourism-related activities and thus opinions about them have been rarely published on the Internet. Therefore, to increase the data volume, we also collected tweets from multiple languages (see Figure 6 for detail of non-English tweets collected). Despite collecting multilingual tweets as well, there were fewer geotagged tweets collected, compared to ungeotagged tweets. See Figure 7 for specific number of tweets collected (geotagged vs. ungeotagged). In fact, only 0.8% of total tweets collected were geotagged. From this observation, we can assume that many tourists do not want to tweet with GPS geotags.

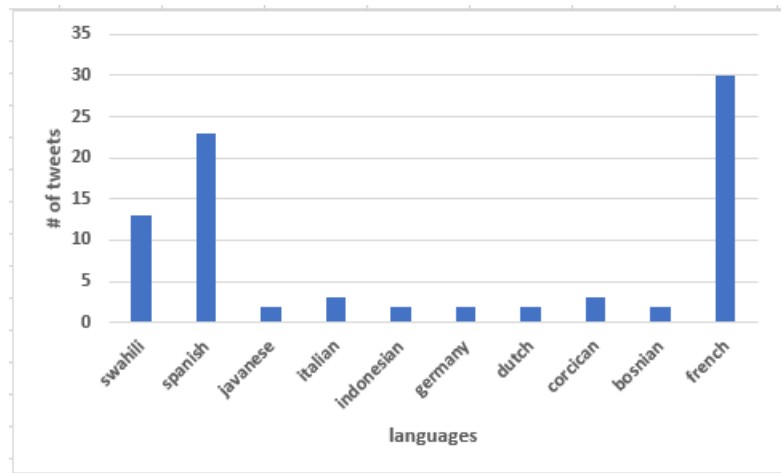

**Figure 6.** Tweets collected in multiple languages.

To bring uniformity among the extracted tweets, the tweets were translated into standard English with the help of automatic Google machine translation service (https://translate.google.com/, accessed on 15 January 2022). All tweets were collected between 3 June 2019 and 24 February 2020 and represent a sample of 155,316 tweets, with 1273 tweets with geotagging information as observed in Figure 7.

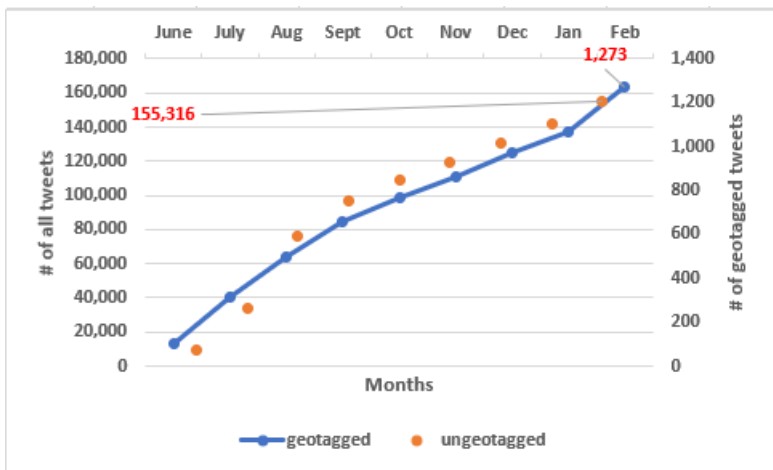

**Figure 7.** Cumulative value of geotagged tweets collected for this research (include multilingual tweets).

## 5. Classification Experiment

*5.1. On-Spot Tweets Detection*

5.1.1. Data

As described in Section 4, a set of 1273 collected geotagged tweets were used in this experiment. After collection, the dataset was prepared in the data pre-processing and feature weighting phase, all tweets were transformed into lowercase. Furthermore, all URLs, e.g., (https://pandasafaris.com/, accessed on 15 January 2022) were removed. This is because the URLs and the tagged users were not likely to contribute to the classification. A traditional weighting scheme was applied to the dataset. In particular, we used term frequency with inverse document frequency (tf*idf) which is used to measure the importance score of words considering frequency of appearing in a document. Therefore, tf*idf is the term frequency multiplied by the inverse document frequency as in the equation below.

$$tfidf(t, d, D) = tf(t, d) \times idf(t, D) \tag{3}$$

where $t$ denotes the terms, $d$ represents document and $D$ denotes collection of documents. We experimentally evaluated the efficacy of the proposed method. At the end of the experiment, we performed a throughout discussion based on experiment results by interpreting the results, evaluating the performance of the proposed approach, pinpointing some challenges encountered, and proposing a way to overcome those challenges.

5.1.2. On-Spot Tweets Detection Using Baseline Classifiers

In this section, we first used baseline classifiers to detect on-spot tweets from a collection of annotated tweets. The dataset used in this experiment consists of 1273 geotagged tweets (974 on-spot and 299 not on-spot) that were manually annotated by three annotators with an inter-annotator agreement calculated with Kappa coefficient as shown in Section 3. Several types of classifiers were applied for comparison in this experiment.

Firstly, a Naïve Bayes classifier was applied. It is a supervised learning algorithm applying Bayes' theorem which assigns class labels to problem instances represented as vectors of feature values and is often applied as a baseline in a text classification task.

Next, we applied the k-Nearest Neighbors (kNN) classifier, which takes as an input k-closest training samples and classifies them based on the majority vote. It is often used as a baseline together with the Naïve Bayes classifier. For the input sample to be assigned to the class of the first nearest neighbor, the k = 1 setting was applied here.

Another used classifier was J48 which is an implementation of the C4.5 Decision Tree algorithm, which builds decision trees from the dataset and the optimal splitting criterion is further chosen from tree nodes to make the decision.

Lastly, Support Vector Machines (SVM) was used. It is a supervised ML algorithm, designed for classification or regression problems, that uses a technique called kernel trick to transform data, and finds an optimal boundary between the possible output. A linear kernel function-based SVM was applied here, as it is known to perform well with text data [26].

Each of the classifiers above was tested on the collected tweet dataset in a 10-fold cross-validation procedure. The results were evaluated using standard Precision (P), Recall (R), and balanced F-score (F1). The results were determined based on the highest achieved balanced F1.

Table 4 shows the summary of the results.

**Table 4.** On-spot detection results obtained from multiple classifiers.

| Classifiers | P | R | F1 |
|---|---|---|---|
| Decision Tree | 0.797 | 0.808 | 0.801 |
| Naïve Bayes | 0.821 | 0.801 | 0.808 |
| KNN | 0.860 | 0.851 | 0.827 |
| SVM | 0.875 | 0.872 | 0.858 |

The results of the SVM classifier were higher than other classifiers. SVM proved to be effective in a binary classification task. This can be because of its effectiveness in high dimensional spaces and also because of using a subset of training points in the decision function (called support vectors), which is also memory efficient.

SVM attained the highest score. Therefore, in the next experiment which we introduce in the next section, we use SVM results as a comparative baseline against the pre-trained BERT model.

The Decision Tree classifier scored the lowest among all the other classifiers. Even though these classifiers may be able to do well in typical sentiment analysis, stemming and parsing are not applied to the dataset as a result of simple data pre-processing; hence, the noisy language might be a challenge for them.

The Naïve Bayes classifier performed slightly better, but close to the Decision Tree classifier.

K-nearest neighbor classifier scored second just after SVM's linear kernel classifier. These results provide important insights into the presence of classifiers in the detection of on-spot tweets while enhancing our understanding of the impact of a training dataset, which is important in the identification task.

In general, ML classifiers demand large volumes of training data to achieve high performance. In this experiment, 1273 geotagged tweets were collected, which was a limited amount of training data. This may have caused the underperformance of some classifiers because they depend a lot on the quantity of training data. In other words, large training data is essential for achieving high results, for this reason it is possible to attain high performance with various classifiers [27].

In this experiment with baseline classifiers, we applied the method to identify on-spot tweets from collected geotagged tweets by building a classifier that uses location clustering and SVM which learns the geotagged tweets information. SVM classifier achieved an average F1 score of 0.858 when compared with other applied classifiers. There was data imbalance, however, only between classification categories in the training dataset, not in test dataset, which suggests that there is a need to collect more data to assure a balance of classification categories in the future study, to improve the reliability of results and decrease potential bias. Table 5 shows the confusion matrix, where we can see 156 instances were incorrectly classified, 120 instances come from "not on-spot" class. Despite the achievement, there is a need to improve the model by collecting more geotagged tweets for training purposes.

**Table 5.** Confusion Matrix from SVM classifier.

| On-Spot | Not | Ref |
|---|---|---|
| 938 | 36 | on-spot |
| 120 | 179 | not |

In the next experiment, we attempted to compare the efficacy of SVM to a deep learning approach.

5.1.3. Baseline vs. BERT

In this section, we compare the efficacy of the SVM classifier to a deep learning approach. To do this, we fine-tuned a pre-trained BERT neural language model and used it for the tweet classification task. Next, we compared the performance of the BERT model with that obtained from SVM. Lastly, we evaluate and discuss the results obtained in this experiment.

In the pre-processing stage, tweets are lowercased. Non-ASCII letters, URLs, @RT: [NAME], are removed. For BERT, texts with a length of less than 4 are discarded. No lemmatization is performed and no punctuation mark is removed since pre-trained embeddings are always used. No stop-word is removed to retain better grasp of the fluency of language.

We demonstrate the efficacy of the deep bidirectionality of BERT by using the same training dataset used as in the previous experiment, with 1273 geotagged tweets.

The original BERT-Base uncased model comprises two models, one with 12 transformer layers, 12 self-attention heads, and the other one with 24 encoders, and 16 bidirectional self-attention heads. Both models pre-trained from unlabeled data extracted from the BookCorpus and English Wikipedia words. In this experiment specifically, we used the distilbert-base-uncasedmodel (https://huggingface.co/distilbert-base-uncased, accessed on 15 January 2022) version of BERT. Compared to other version of BERT models, a DistilBERT is significantly smaller, consistently faster, while retaining high performance when compared to original BERT model [28]. Training neural language models from scratch is typically time consuming. Even fine-tuning the pre-trained model with a task-specific dataset may take several hours to finish one epoch, as shown by Padigela et al. [29]. Thus, in reducing computational time in this experiment, we deploy a ktrain library (https://github.com/amaiya/ktrain, accessed on 15 January 2022) which is a lightweight wrapper for tf.keras in TensorFlow 2. It is designed to make the deep learning process more accessible and easier to apply, as described by Maiya et al. [30].

We, therefore, train our model in consecutive 2-epochs. Distilbert-base-uncased model is trained using the same corpus as the original BERT model which includes a concatenation of English Wikipedia and a book corpus in a self-supervised fashion using the BERT base model as a teacher.

BERT performed well with the same amount of data for the on-spot review tweet identification task when compared to SVM. Table 6 demonstrates the obtained results. The reason for such a high score was most probably due to BERT working best with the context orientation of the sentence, hence simplifying the classification task. This observation suggests that a deep learning approach can show a significant improvement when dealing with limited training data. Pre-trained language models such as BERT have proven to be highly effective for NLP tasks. However, the high demand for computing resources in training such models from scratch hinders their application in practice.

**Table 6.** Classification results obtained from BERT model with comparison to SVM.

| Model | P | R | F1 |
|---|---|---|---|
| SVM | 0.875 | 0.872 | 0.858 |
| BERT | 0.927 | 0.946 | 0.936 |

### 5.2. Adding Rating Score

In addition to classifying the tweets as candidates for on-spot reviews, we also needed to assign rating scores to the tweets containing the opinions about the sight spots, as knowing that the tweet contains an opinion is not sufficient to make it usable in practice. We also needed to know what is the semantic polarity of the opinion, or, whether the opinion is positive, negative, or neutral (subjective, but not loaded description of the sight spot).

#### 5.2.1. Data

We only used on-spot judged tweets as a data input for this experiment (on-spot judged tweets = 974 tweets). This is because, on-spot judged tweets with their correspondence sentiments score were used in the designed system (PSRS), (see the distribution of annotated tweets in Table 2).

#### 5.2.2. Model

To do that, we applied the sentiment annotations assigned during the annotation process to the extracted tweets and again trained BERT classifier automatically assign most probable rating information.

With three classes (positive, negative, neutral), this became a multi-class classification problem and to accomplish this task we tested two separate class intervals. One with a 3-classes range and another one with 5-classes range.

#### 5.2.3. Annotation with 5-Star Rating Interval

We set a 5-score range and annotated tweets using three people whom each annotated as follows.

- 5 star—for a very positive opinion
- 4 star—for a positive opinion
- 3 star—for a neutral opinion
- 2 star—for a somewhat disappointing opinion
- 1 star—for a harsh or disappointing opinion

After annotation, we decided on the rating information by taking the average score between three annotators for each specific tweet. Moreover, if a tweet received the same score from two different annotators, we used that annotated score.

After the annotation, 5 tweets were annotated as 1 star, 29 tweets for 2 stars, 276 tweets for 3 stars, 440 tweets for 4 stars, and 224 tweets for 5 stars, as observed in Table 7.

**Table 7.** Annotation summary—5 scale range.

| Rating | 5 | 4 | 3 | 2 | 1 |
|---|---|---|---|---|---|
| Tweets Counts | 224 | 440 | 276 | 29 | 5 |

#### 5.2.4. Annotation with 3-Star Rating Interval

In contrary to the first five-score range, we also set a separate group with a three-range score. The aim is to experiment with both intervals and compare the results between these two different scores range. Here, the tweets' score was grouped as follows.

- 3 star—for a positive, impressive tour experience sentiment (5 and 4)
- 2 star—for a neutral sentiment (3)
- 1 star—for the somehow disappointing or harsh sentiment (1 and 2)

After annotation, the results were as observed in the Table 8, namely, 34 tweets for a 1-star rating, 276 tweets were annotated with a 2-star rating and 664 tweets were annotated with a 3-star rating. Table 9 shows a few examples of annotated tweets with added score information.

**Table 8.** Annotation summary—3 scale range.

| Rating | 3 | 2 | 1 |
|---|---|---|---|
| Tweets Counts | 664 | 276 | 34 |

**Table 9.** Examples of star-rated annotated tweets.

| Tweet Contents | POI | Score |
|---|---|---|
| amazing elephant experience with #oliviatravel today in the #serengeti #grateful #elephants #wildlife at four season | four seasons safari lodge | 3 |
| ngorongoro crater at ngorongoro national park | ngorongoro crater | 2 |
| we enjoyed our day around lake ndutu and serengeti#safari #safaritanzania#tanzania#tanzaniasafari#landcruiser | lake ndutu | 3 |
| worse places to get some writing done #amwritingscifi#tanzania #travel #writersofinstagram at kiota camp serenget | kiota camp | 1 |

## 6. Results and Discussion

We adopted BERT for on-spot tweets classification and sentimental polarity prediction. The results show BERT outperform baseline classifiers in binary classification task. In the sentiment polarity classification, Table 10 shows the prediction performance was better for the 3-star range interval compared to 5-star score range. A three-score range setup outperformed a five-score range scale with an F-score of 0.74. 5-star and 3-star, and the smaller number of classes results in more samples per each class and eventually allows for better generalization of data. Recently, (Kayastha et al. [31]) demonstrated a procedure to tackle class imbalance by addition of per- class weights to the standard cross-entropy loss function, which shows better results compared to oversampling or undersampling. Therefore, it will be our consideration for future improvements.

**Table 10.** Summary of score assigning results.

| Score Range | Accuracy | F1 |
|---|---|---|
| 5-star score range | 0.69 | 0.66 |
| 3-star score range | 0.77 | 0.74 |

On the other hand, as observed in Table 11 there was classifier misjudgement between annotated score and predicted score. We identified these tweets as difficult to judge. Figure 8 also shows our model evaluated a negative sentiment tweet (tweet number 23) as positive sentiment. One way to improve our model performance is to remove tweets with high degree of ambiguities in training set. This will be our consideration in our future works.

The results demonstrated that our proposed method, although not ideal, is sufficiently usable to be used for score generation.

**Table 11.** Examples of misjudged tweets.

| Tweet Contents | Annotation | Prediction |
|---|---|---|
| we spent our final safari day at ngorongoro crater it was surprisingly cold but we had a rare chance of [...] | 1-star | 3-star |
| there is no #wifi on a #safari but youll find a better connection #tanzania #ngorongoro at ngorongoro | 1-star | 3-star |

Moreover, wildlife-related sentiments differed significantly. For example,

- Serengeti is basically just animals killing one another
- I have been to Africa and the Serengeti. I have seen hundreds of giraffes. Killing one as a sport
- Serengeti: pride of lions hunting and killing zebras
- #serengeti That lion killing the cub, has put me in such mood. i'm absolutely livid

Words such as "animal killing", "killing" can be perceived as dangerous, scary which could potentially cause their lower rating generation. This contextual ambiguity poses a challenge in the automatic prediction of wildlife sentiment rating. To deal with this it is necessary to remove noisy data, hence improve degree of agreement between annotators.

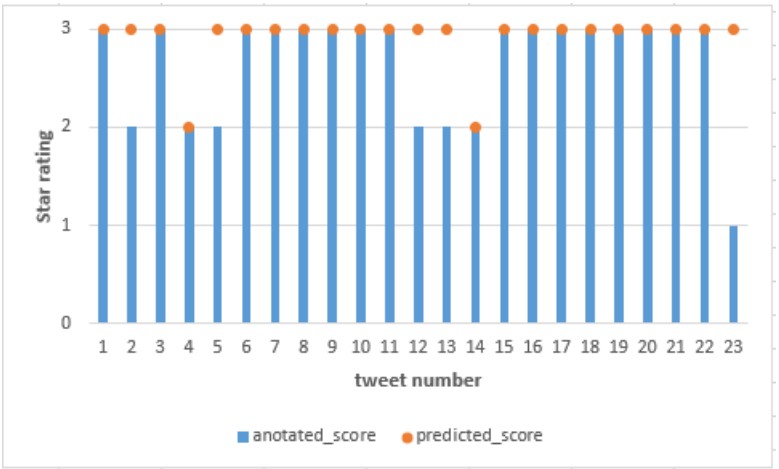

**Figure 8.** BERT score prediction results for a 3-class range.

### 7. PSRS

Tourism, for many areas in the world, is one of the most important industries. The activation of tourism leads to the activation of industries and communities. In this situation, easy access to information provided on the World Wide Web plays an important role. In this study, apart from developing two methods—one for detecting opinionated tweets, and second for assigning sentiment score, we also designed a system (PSRS) on which we mapped all on-spot judged tweets (on-spot reviews) as tourist's opinions.

Figure 9 shows the interface of the designed system in a mobile view. The system displays verified tweets (on-spot reviews) as tourist's opinions collected from the target spot using our method of complementing the lack of reviews for the rarely visited sightspots. Figure 10 shows the overview of the designed system. In this system, user can search and select registered sightspots and new POI (see Figure 11 for example of sightspots registered in the system). The database of our designed system holds about 165 sightspots extracted as sightspots and POI, respectively. Extracted sightspots include, wildlife spots, accommodation spots, souvenir spots and restaurant spots in the target spot. Upon selection, the system does not only display the verified tweets (on-spot reviews) corresponding to the selected sightspot, but also displays rating information added on the sightspot. Optimal routes between two points in the target spot can also be displayed (see Figure 12). We used Google Maps API for this function. Since one sightspot possesses number of rating scores evaluated from multiple on-spot reviews, respectively, we also compute the average rating score of a given sightspot by using the formula below.

$$Sight_{scr} = \frac{1}{n} \sum_{i=1}^{n} scri \tag{4}$$

- $Sight_{scr}$ = Sightspot score
- $n$ = number of score items
- $scri$ = the value of each individual score given

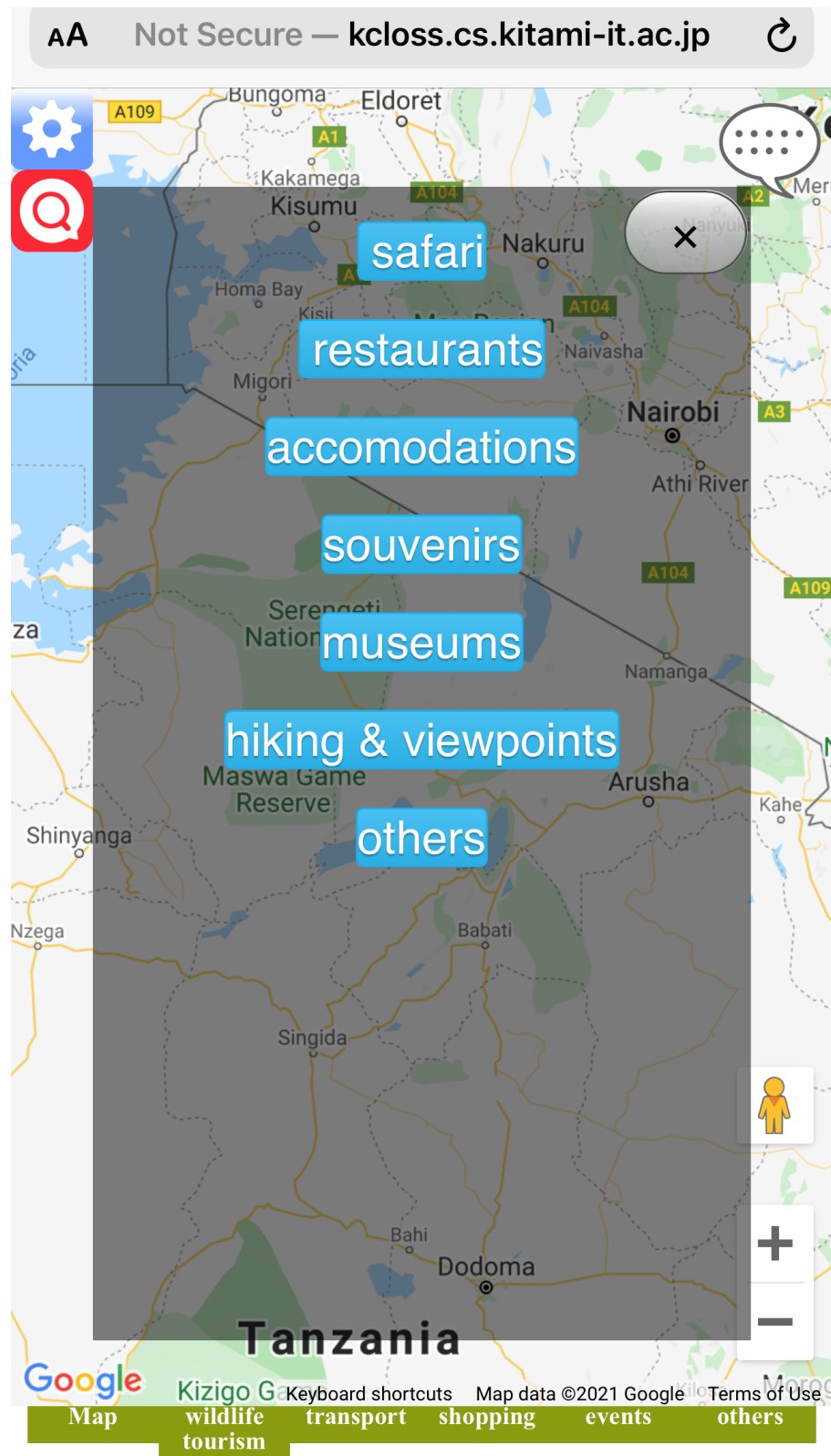

**Figure 9.** System interface (mobile view).

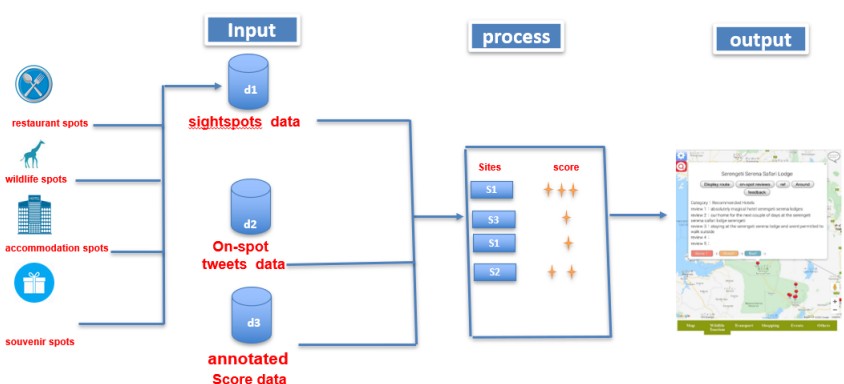

**Figure 10.** System overview.

Further use the computed score to represent the sightspot rating information (see Figure 13).

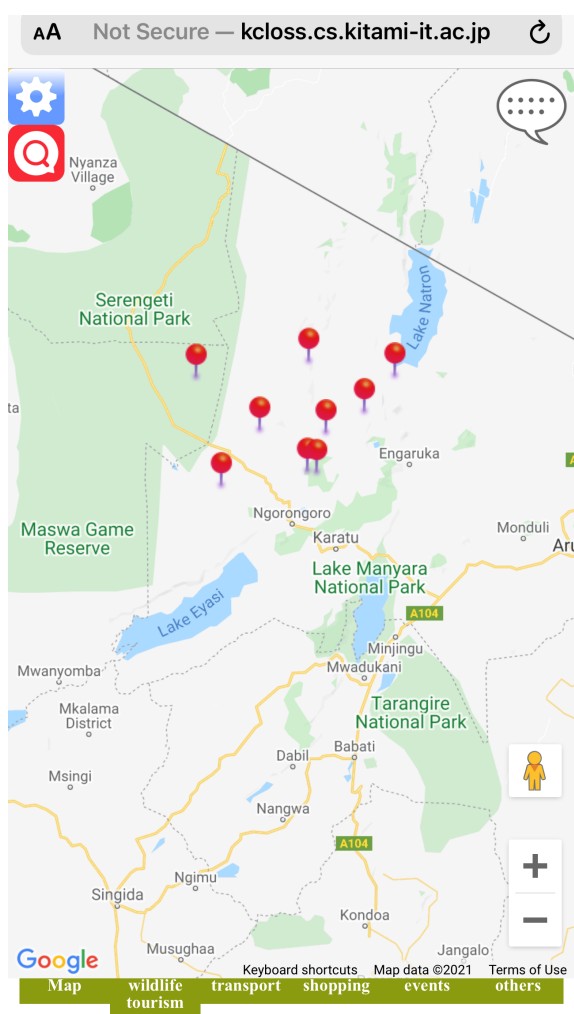

**Figure 11.** example of registered sightspots in the system.

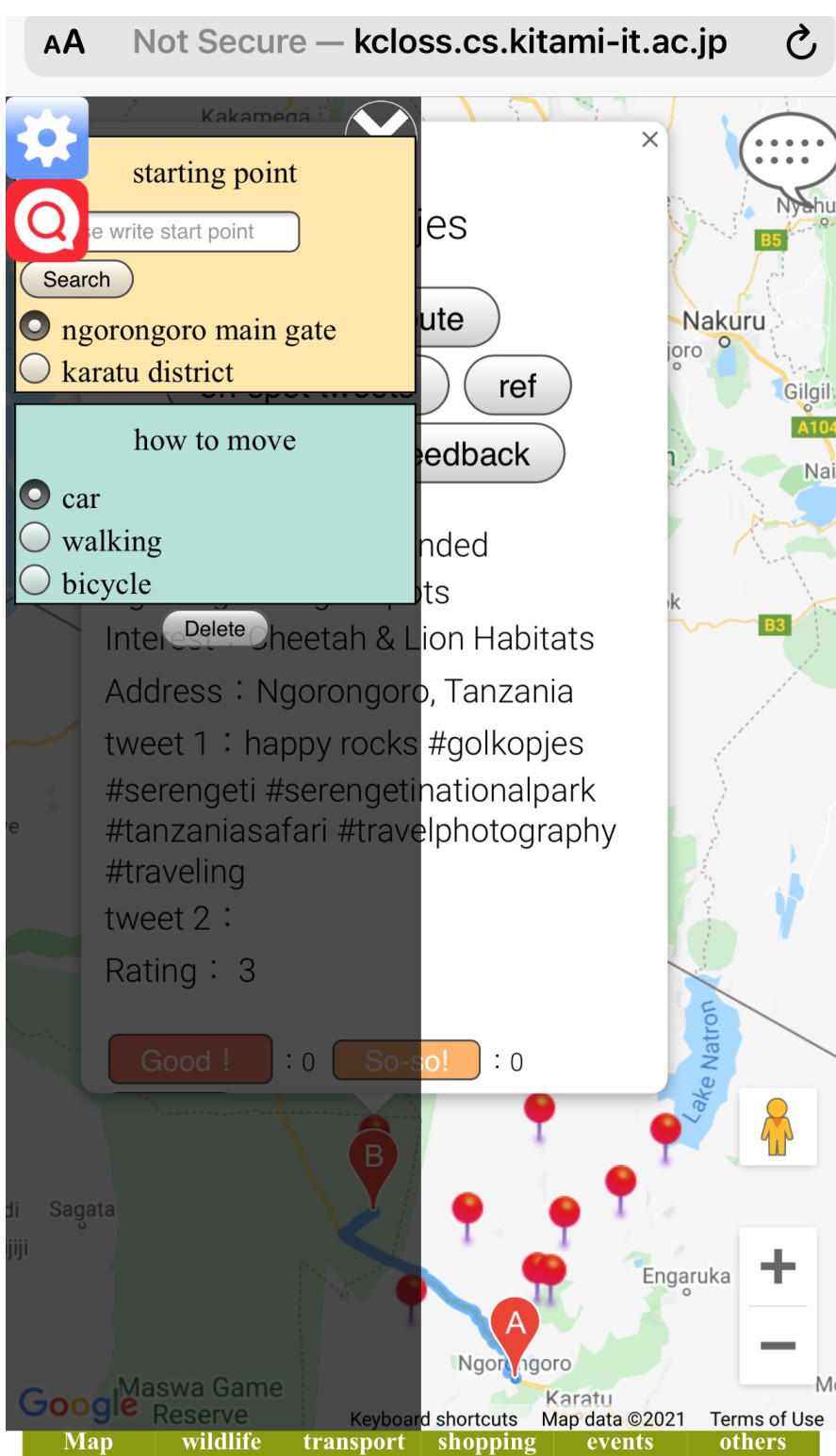

**Figure 12.** Route search display.

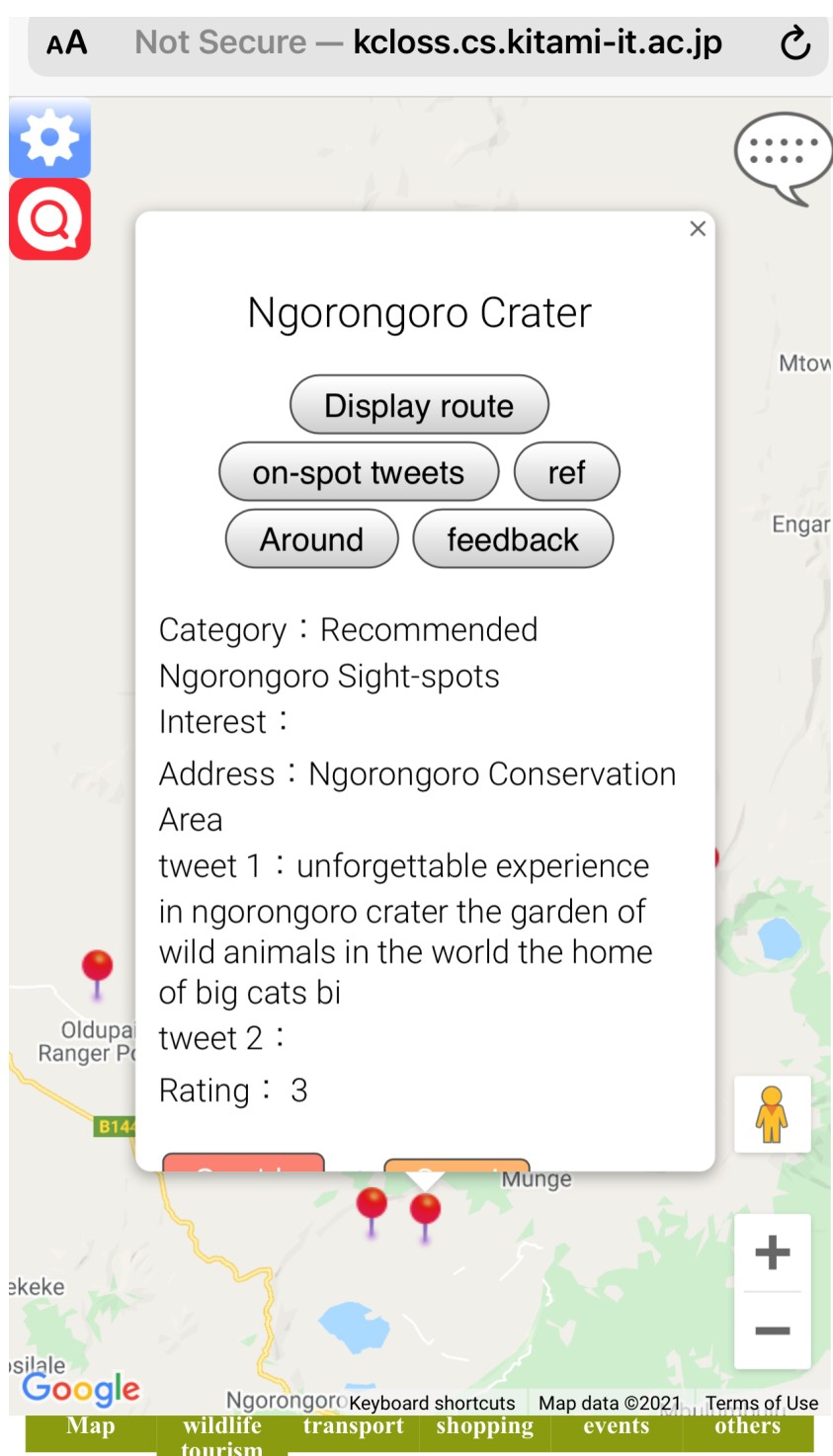

**Figure 13.** Rating information on sightspot .

The designed system intends to support local tourism in Tanzania specifically in the area of wildlife-base tourism.

*POI Discovered*

Location and time data associated with extracted tweets can be considered as useful geographically annotated materials on the Web. They generate detailed tourist trails of which regions have been visited more or are more attractive (Lee et al. [32]).

Figure 14, visualize the distribution of geotagged tweets collected, some in the target spot (possible on-spot tweets) and others outside the target spot (not on-spot). From this

figure, we can observe tweets posted in different areas (here referred to as sightspots) with precise location and time of the day. This helps us analyze the posting behavior and extract useful information such as POI, events, trends, or activities. For example, we can observe late-night tweets at some points in the target spot which can possibly represents on-spot reviews of the accommodation facilities used by the tourists. Another example is seen from the accumulation of tweets from the same location which represents the most visited spots, (For example, in Figure 15 shows 92 times for ngorongoro crater spot). This information is also useful in recommending sightspots, or sightspot routes.

Table 12 shows few examples of tweets with newly found POI.From our analysis, we discovered 68 different POI. Figure 15 illustrates a bar graph that shows sightspots discovered with the number of occurrences in tweets. For example, "ngorongoro crater" has the largest number of on-spot tweets (92 tweets). It shows this sightspot was the most visited. We were interested to cultivate such information as season and time of the day. Figure 16 shows the visiting progression in a year basis and time of the day.We can learn the time interval between 15 through 18 accumulates most of the tweets, which can possibly represent the preferable crater visiting time. Another example of point of attraction discovered is the Mara River. An annual scenery of wildebeest migration in the planes of Serengeti happens across this river. We can identify different observation points throughout the river for best scenery using geolocation information tagged in the tweets. For example, Figure 17 shows a point of intersection between the river channel and sightseeing pattern which can possibly represent the optimal wildebeest migration view point.

**Table 12.** Examples of tweets with newly found POI.

| Tweet | Created | Time | Longitude | Latitude | POI |
|---|---|---|---|---|---|
| had a chance of lifetime to see this rhinoceros in the ngorongoro crater i will be making a present | 12 June 2019 | 18:44 | 35.6762 | −3.1540 | ngorongoro crater |
| ngorongoro crater hippo pool @ ngorongoro crater | 17 June 2019 | 10:09 | 35.6762 | −3.1540 | hippo pool |
| successfully completed a baloon safari in central serengeti seronera | 18 June 2019 | 23:35 | 36.6833 | −3.3666 | seronera |
| sunrise and assortment of breakfast delights last day at camp oleserai@ oleserai luxury camp | 9 June 2019 | 5:28 | 34.7521 | −2.6105 | oleserai luxury camp |

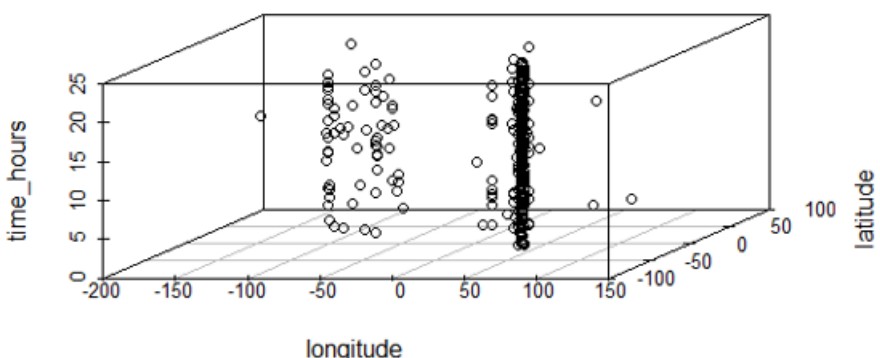

**Figure 14.** Distribution of geotagged tweets collected.

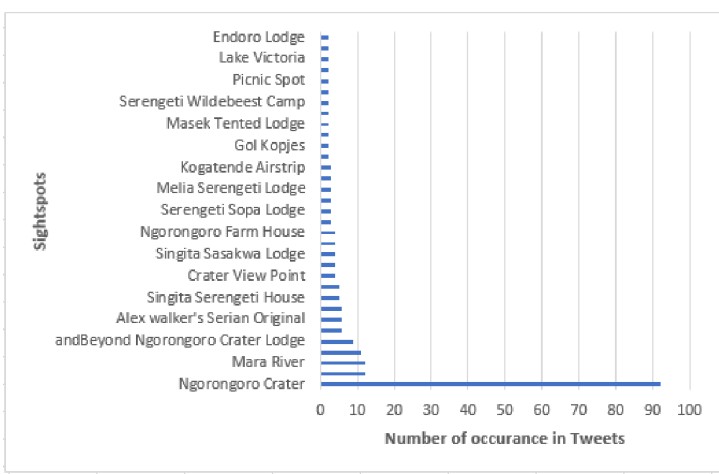

**Figure 15.** POI discovered.

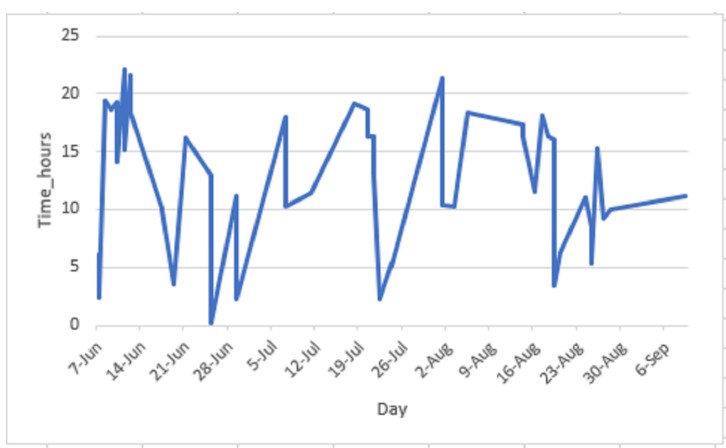

**Figure 16.** Crater visiting time.

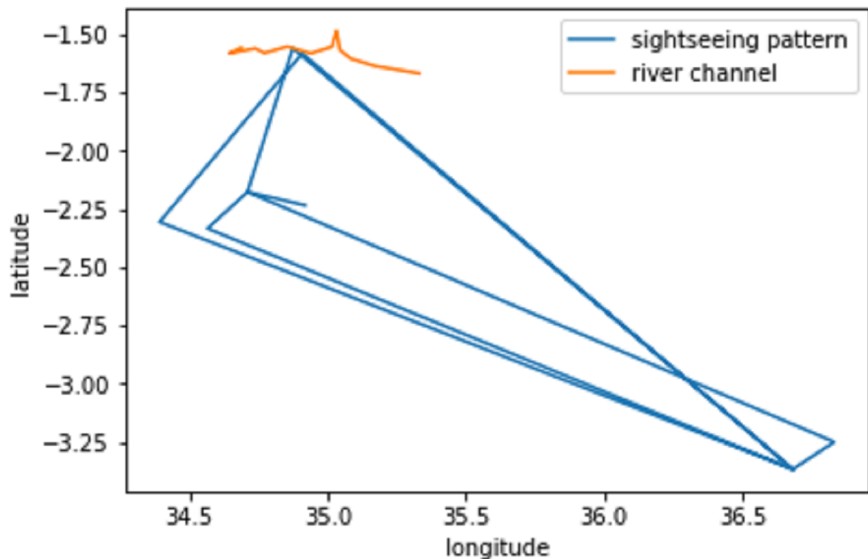

**Figure 17.** Observed wildebeest migration viewing pattern.

## 8. Conclusions

In this study, we proposed a new method to extract from the Internet new on-spot tourist opinions for the tourism information analysis system, by collecting Twitter data

and building a classifier that distinguishes on-spot tweets from a set of collected tweets and automatically adds rating information to the opinion by using a BERT neural language model-based classifier which learns the geotagged tweets information.

The proposed method incorporates a location clustering and classification technique using multiple algorithms including state-of-the-art neural architecture. In the experiment, we used location-clustered tweets to build a classifier that learns information from on-spot annotated tweets to further classify those tweets and compare various classifiers. The results showed that the best performance was achieved by the baseline (SVM) classifier which achieved a high F-score of 0.85 compared to others.

Finally, we compared the SVM classifier to a deep learning state-of-the-art technique (BERT), utilizing the same tweet dataset. Experiments showed that BERT outperformed SVM and achieved a high F-score of 0.94. Despite its demand for high computing power, BERT showed excellent results with only limited training data. It suggests that a BERT model can be adopted in solving the task of on-spot tweets identification and sentimental polarity prediction in particular when there is a challenge of limited training data.

Classified on-spot tweets with their added rating information were mapped as on-spot reviews into the designed system (PSRS) as sightseer's supplementary opinions. From the classified on-spot reviews, we also took efforts to discover POI from the tweets and present them as interesting sightseeing points.

Since we built a classifier that automatically detects on-spot tweets and adds rating information to them by solely relying on geotagged tweets, it would be interesting to use this classifier to predict also non-geotagged tweets. Therefore, we will consider that in our future studies. Furthermore, we hope our corpus (Table 2) of on-spot annotated tweets can be used in the future for the deployment of prediction system.To increase the usefulness we also plan to increase the data volume of this corpus. We hope that this study will inform and enrich other researchers and would be useful for future studies on also exploring the application of NLP, Big Data, and Artificial Intelligence to the full advantage of the revitalization of regional tourism in areas other than Tanzania.

**Author Contributions:** Conceptualization, V.S., F.M. and M.P.; methodology, V.S. and F.M.; software, V.S.; validation, F.M.; formal analysis, V.S.; investigation, V.S.; resources, F.M.; data curation, V.S.; writing—original draft preparation, V.S.; writing—review and editing, M.P. and F.M.; visualization, V.S.; supervision, F.M. and M.P.; project administration, F.M. and M.P. All authors have read and agreed to the published version of the manuscript.

**Funding:** This research received no external funding.

**Institutional Review Board Statement:** Not applicable.

**Informed Consent Statement:** Not applicable.

**Data Availability Statement:** Not applicable.

**Acknowledgments:** The authors would like to thank the anonymous reviewers for their constructive feedback.

**Conflicts of Interest:** The authors declare no conflict of interest.

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
