# Peer review of "A Method of Supplementing Reviews to Less-Known Tourist Spots Using Geotagged Tweets"

_applsci, doi:10.3390/app12052321_

Round 1
Reviewer 1 Report
I hope the authors find my comments productive and that they will help them improve their research work.
In this paper the authors investigate a method of supplementing reviews to unpopular tourist spots using geotagged tweets. The keywords match the terms used in the research. The Introduction section puts the reader in context but the authors are asked to improve this section and to elaborate on the relevant and up-to-date research topic related to the topic. The objective of the research is clear and well defined.
The structure of the article is clear. Materials and methods are adequately described.
Results and discussion - In this section, the authors should comment on the results obtained by referring to previous studies, thus confirming what other research has shown. They should also highlight the limitations of the research carried out and propose new lines of research on the object of study – please correct this section.
Conclusions - With regard to the conclusions, the authors must clearly state the conclusions that have been obtained from the research. Similarly, it is advisable to refer to the objectives set out in the research and the findings or results obtained, thus giving an overall view of the whole study.
References are few (28) of which more than 80% are up to date – please add more actual references.
Reviewer 2 Report
The paper "A METHOD OF SUPPLEMENTING REVIEWS TO UNPOPULAR TOURIST SPOTS USING GEOTAGGED TWEETS" is an interesting and well-written paper presenting a novel solution to a problem of poor availability of online reviews to some dispersed tourist spots. The procedure is innovative and well-conceived, the experiment is well designed and the practical application of the system is potentially useful. Therefore I recommend the paper for publication. There are only a couple of minor editorial issues that I would like to point out:
Lines 52 and 78 - reference number is missing.
Starting from line 155 - subsections should be numbered rather 2.1 etc. than 2.0.1 etc.
Line 562 - what does "WB" abbreviation stand for?
Reviewer 3 Report
Very interesting approach.
Looking forward to your future research.
Reviewer 4 Report
Although valid data represent a small percentage of those obtained, the method used seems interesting and perhaps with possibilities of being applied in other places and with different circumstances.
Regarding the structure, the title of the figures would have to be improved and that it appears before and not after them.
In addition, it is also observed that the position in which the interpretation of the content of some tables appears does not appear next to them, which complicates their follow-up a little.
Also related to the figures, some do not appear as No. 11 and the position of others must also be corrected because many blank spaces appear around them, along with the size they have it is difficult to appreciate what they collect.
